# Progressive Coarse-graining and Deep Neural Networks (DNNs)

## Abstract

We try to provide an overarching perspective on some of the research done in the last few years explaining the behaviour of deep neural networks (DNNs) when they are used to complete a variety of classification and prediction tasks. We start by providing an overview of several noteworthy papers on the fundamental properties of DNNs across different architectures and data regimes. We then forward our own integrated perspective of DNNs as progressive coarse-graining systems inspired by Erik Hoel's Causal Emergence 2.0 framework.

## 1 Introduction

Deep Neural Networks (DNNs) performing supervised learning (SL) have shown impressive performance in a range of prediction and classification tasks. Despite this, relatively little information is known about why and how they are capable at these tasks.

While empirical work has been carried out to estimate the capabilities of networks of various sizes for Large Language Models (LLMs, see Kaplan et al. (2020), and mechanistic interpretability work has been carried out for a range of models and architectures (see Olah et al. (2020) and later work like Lindsey et al. (2025)), a high level explanation behind the mechanisms of SL models is still lacking. We propose a potential high level explanation of how and why overparameterised DNN models succeed at classification and prediction tasks, inspired by Erik Hoel's Causal Emergence 2.0 Framework Hoel (2025).

To verify this model, we also conduct a literature review over past work on DNN learning behaviour Feng et al. (2022), including research on the rank of DNN subnetworks, hierarchies of concepts in transformers Dorszewski et al. (2025), Tishby's Information Bottleneck theory (See Shwartz-Ziv & Tishby (2017)) and the Lottery Ticket Hypothesis (See Frankle & Carbin (2019)). We find strong evidence from multiple sources that neural network layers perform progressive simplification of input data in service of prediction and classification, discarding irrelevant information and exposing relevant information. Our work contrasts against circuit-based interpretability work, which is limited to particular case studies and isolated behaviours, by offering a general picture of DNN behaviour after training via stochastic gradient descent (SGD).

## 2 Definitions

A **deep neural network** (DNN) is a multi-layer perceptron (MLP) with up to $L$ layers. At each layer $i$ the output is given by the formula:

$$x_i = \text{act}(w_i x_{i-1} + b_i) \tag{1}$$

Where $w_i$ is the weight matrix, $b_i$ is a bias vector, and act is a nonlinear activation function e.g. ReLU (except for the last layer, when act is the identity function). The input is $x_0$ or just $x$. The output is $x_n$, usually denoted $\hat{y}$ to separate it from the output label $y$.

Our convention is to label a neural network $F$ and the function at layer $i$ of that network $f_i$. The subnetwork formed from layer 1 to $i$ of a network $F$ is written as $F_{1:i} = f_i \circ f_{i-1} ... \circ f_1$. Here $\circ$ is the composition operator.

A **feature** or **feature vector** is a column in a weight matrix $w_i$, which we denote as $w_{i,j}$. The formula for the $j$th value in the output vector $x_i$ is shown here to make the relationship between features and output explicit:

$$x_{i,j} = \text{act}(w_{i,j}x_{i-1} + b_{i,j}) \qquad (2)$$

A **feature space** is the space formed by linear combinations of all features in a given weight matrix multiplied by some scalar. This space may not be a valid vector space because features in any given weight matrix are not guaranteed to be linearly independent.

The **rank** of the feature space of a layer $f_i$ (denoted $\text{Rank}(f_i)$) is analogous to the rank of the weight matrix. The **intrinsic dimension** of a layer is denoted $\text{Dim}(f_i)$.

The **numerical rank** of a function $f$ (denoted $\text{Rank}_{(}\text{num})(f)$) is taken from the definition given in Feng et al. (2022), where they use it to measure the approximate rank of DNN layers. A function's numerical rank is defined as the rank of its Jacobian matrix over some input domain. When we are estimating the numerical rank of DNN layers, this is troublesome because even a small amount of noise can make an otherwise very low-rank matrix full rank. To get around this the authors estimate the numerical rank by counting the number of non-zero values in the singular matrix $sigma$ of the Jacobian matrix. To correct for noise, the singular values must be higher than a threshold $epsilon||W||_2$ to be counted. Here $||W||_2$ is the L2 or spectral norm of the Jacobian matrix and $epsilon$ is a fixed error threshold.

The **effective rank** of a layer $f_i$ (denoted $\text{Rank}_{\text{eff}}(f_i)$) of a network $F$ with layers $1...i...L$ is the numerical rank of the function $F_{1:i-1}$ Similarly, the **effective feature space** of a layer $f_i$ is the space formed from the possible outputs of $F_{1:i-1}$ and the feature space of $f_i$ [1]

When we mention the rank or feature space of a layer in a DNN, unless specified otherwise we always mean the effective rank or the effective feature space of that layer. We further note that as a rule in DNNs $\text{Rank}_{\text{eff}}(f_i) < \text{Rank}_{\text{num}}(f_i)$ and the effective feature space is smaller than the feature space.

## 3 PRIOR WORK

A lot of work has been done in neural networks to study exactly how they process inputs. According to Shi et al. (2025), "Deep neural networks (DNNs) progressively compute features from which the final layer generates predictions. When optimized via stochastic dynamics over a data-dependent energy, each layer learns to compute better features than the previous one, ultimately transforming the data to a regular low-dimensional geometry." According to Xu et al. (2020), DNNs work by "extracting hierarchies of progressively more informative features in representation learning". The idea of *progressivity*, i.e. that each layer improves upon the performance of the previous layer, is key.

We add to that the narrative given in Feng et al. (2022). It says that as you progressively compute features layer by layer, the numerical rank of each layer also decreases monotonically and progressively. This is intrinsically connected to the rank of the feature space (weight matrix), because each layer consists of multiplying the input by the weight matrix and then applying some nonlinear function.

They then show that the fundamental primitives of neural networks (namely, matrix multiplication, the chain rule, and nonlinear activation functions) can only construct functions whose numerical rank is equal to or lower than the numerical rank of the functions that serve as their inputs. In other words, a neural network is not only incentivised to project input data into a "regular low-dimensional geometry", but is in fact equipped with an inductive bias that requires such a projection. This is true even if the width of the hidden layers remains stable.

---

[1]Consider the case where $f_i$ has 64 feature vectors. Each feature vector has 6 values and are linearly independent, but they differ in only the 6th value. So one vector might be $[0, 1, 5, 2, 3, 6]$ and another might be $[0, 1, 5, 2, 3, 7]$. Now suppose that the output of $F_{1:i-1}$ is always 0 in the sixth value. Effectively, the 64 feature vectors are now identical, making the effective feature space much smaller!]

Such an inductive bias raises the possibility of *oversimplification*, since a DNN is *required* to decompose the input into lower-rank feature spaces by the nature of its construction. If the feature space becomes too "rank deficient", however, the authors show that many of the terminal features in a DNN can be modelled as linear combinations of other terminal features. This is true even if the features themselves are semantically unrelated. In a particularly striking example, the output strength for the "junco" bird category in the GluMixer-24 model can be effectively predicted by simply multiplying the strength of the "triumphal arch" building category by a fixed coefficient $(-0.923)$.

Next, we note the work of He & Su (2023). Effectively, the authors show that as you go down the layers of a DNN, the resulting features can be mapped to unique target classes via linear regression more and more clearly. This separation is smooth and regular, and increases progressively with each layer just as the rank decreases progressively with each layer. This phenomenon on its surface contradicts computation or circuit-based models of DNN functionality which are generally based around the idea of computing non-smooth logical operations with features (e.g. taking the logical AND of two features to form a composite feature).

In an information theory setting, we can treat each layer like a signal-generating function. In that case, rank diminishing can be correlated with a decrease in the intrinsic dimension of that function (this is also noted by the authors of Feng et al. (2022)). This indicates that fewer independent variables are required to represent the function's behaviour, meaning that the function contains less information overall. The increasingly low numerical rank of these functions (down to possibly rank-1) highlighted in the rank diminishing research also suggests that information is being discarded at each layer between the input and the output. This means that more elements of the output will be dependent each other compared to the input, which is what happens when inputs are multiplied by a weight matrix containing many redundant features. This information, once lost, cannot be easily recovered: we usually cannot construct a true inverse for a DNN layer which involves a matrix multiplication and a nonlinear activation [2].

Finally, the comparison of neural networks with Bayesian modelling has been an ongoing field of study, leading to work like Bayesian neural networks (see Yu et al.) and Bayesian RNNs (see Coscia et al. (2025)). In this regard the work of Mingard et al. is particularly significant. In Mingard et al. (2020), the authors point out that DNNs trained by SGD effectively approximate what Bayesian learning suggests should be ideal function $f$ for predicting some set of observations $S$. They also point out that the Bayesian estimation process strongly prefers simple "low-error and low complexity functions". Taking into account the rank diminishing phenomenon, we believe this means that the DNN's inductive bias towards simplification and low-rank feature spaces is in fact beneficial in most cases. The high compatibility between SGD and Bayesian estimation implies that the "ideal" Bayesian posterior functions produced by Gaussian estimation also ignore large amounts of irrelevant information present in the input and are hence "simple" in an information-theoretic sense.

## 4 DNNs AS PROGRESSIVE COARSE-GRAINING SYSTEMS

Here we aim to present an overarching perspective on how DNNs achieve strong performance. We emphasise the similarities that arise between different frames in light of the prior work we have examined. Overall, A deep neural network can be described as a principled simplification process with the goal of procedurally removing irrelevant information from the input, resulting in regular and predictive features that can be used to minimise loss.

### 4.1 WHAT IS COARSE-GRAINING?

Returning to our initial problem of understanding the mechanics of DNNs, we can see DNNs then as a means of progressively coarse-graining inputs layer by layer. By coarse-graining we mean discarding microscale low-level information in a manner which improves macroscale high-level predictive performance, similar to the definition offered by Hoel (2025) [3]. For example, an image

---

[2]This is trivially true especially with a activation function like ReLU, which simply zeroes out the negative components of any input.

[3]However, we do not follow Hoel in requiring that the resulting simplified system demonstrate the same behaviour under random walkers, since we are not analysing markov chains.

input might to a human viewer contain the information "this is an image of a cat with orange fur and a black tail sitting on top of a brown cabinet". After coarse-graining only prediction-relevant high-level information remains—for an image classification task, that information is "this is an image of a cat".

Importantly, this process cannot be trivially reversed. We cannot get a precise image of any given cat back just by being told the probability that an image is of a cat. The idea that DNNs are coarse-graining, then, is correlated with the information loss at each layer we explored with regards to Feng et al. (2022), and the correlation of progressive information loss with progressively improving classification performance is validated by He & Su (2023). From this lens, we can say that the primary operation of the DNN is to simultaneously reduce extraneous information in the input and expose prediction relevant information [4].

The progressive coarse-graining perspective also generalises the argument put forth in Olah et al. (2020) and related works where CNNs are depicted as having a hierarchy of features. This hierarchy is not only about the predictive quality of the features (as proposed in Xu et al. (2020)), but about semantic qualities of the features. The hierarchy goes from very simple, generic, and low-level features like edge or shape detectors at the first few layers to very complex, specific, and high-level features like the face of a dog or a car door at the later layers. The multiple layers of the hierarchy are important, since if a high-level concept or terminal feature is identified from very early layers it would disrupt the progressive coarse-graining narrative. However, in CNNs this hierarchy is explicitly encoded via the use of kernels, and does not offer much evidence for progressive coarse-graining being present in DNNs without such built-in modules. Our perspective suggests that a similar hierarchy of features from low-level to high-level must be present in DNNs as well. This should be true even when the size of the hidden layer is fixed and there is no inductive bias towards the concatenation of local information created by the use of kernel and pooling operations.

Such a hypothesised hierarchy of concepts has since been demonstrated by Dorszewski et al. (2025). In that paper they show that a hierarchy of concepts from low-level and generic ideas like "colour" and "shape" to high-level and specific ones like particular objects and creatures is present in Vision Transformers. They detect this hierarchy by measuring the strength of simple and complex concepts across MLP layers with neuron labelling. We note specifically that Vision Transformers have uniform residual stream length across all layers and do not use CNN-style kernels in between MLP layers [5]. Therefore, the existence of similar conceptual hierarchies in Vision Transformers and CNNs is significant and shows that progressive coarse-graining is present across different architectures and independent of encoded inductive biases.

## 4.2 Coarse-graining and Information Bottleneck theory

We see some more validation of this argument when we analyse DNN behaviour through the lens of Tishby et al.'s Information Bottleneck framework. As established by Shwartz-Ziv & Tishby (2017) and confirmed in Goldfeld et al. (2019), early layers of trained neural networks consistently show higher mutual information with inputs compared to latter layers, which privilege mutual information with ideal outputs. This gradient from relatively high to relatively low mutual information is, again, monotonic and progressive. This suggests that DNNs are in fact performing progressive coarse graining, removing irrelevant information from the input data layer by layer such that what remains is information that may not be very representative of the inputs but is much more predictive of the correct outputs.

---

[4]For a more information-theoretic framing of this process, see Appendix I.

[5]Vision Transformers (ViT) ingest images as patches, using the attention layer to share information across patches and the MLP layer to aggregate or modify information within patches. The authors of the Vision Transformer paper specify that there are very few image-specific or hierarchical inductive biases encoded in the architecture: *"We note that Vision Transformer has much less image-specific inductive bias than CNNs. In CNNs, locality, two-dimensional neighborhood structure, and translation equivariance are baked into each layer throughout the whole model. In ViT, only MLP layers are local and translationally equivariant, while the self-attention layers are global. The two-dimensional neighborhood structure is used very sparingly: in the beginning of the model by cutting the image into patches and at fine-tuning time for adjusting the position embeddings for images of different resolution [...] Other than that, the position embeddings at initialization time carry no information about the 2D positions of the patches and all spatial relations between the patches have to be learned from scratch."* From Dosovitskiy et al. (2021), emphasis mine.

Butakov et al. (2024) extend this work by showing that training of DNNs has two distinct stages consistent with the Information Bottleneck theory: a smooth "fitting" stage, where intermediate representations gain mutual information with both inputs and outputs, and then a "compression" stage, where mutual information between intermediate representations inputs across the layers is sacrificed in favour of greater mutual information with (i.e. more effective prediction of) outputs. This is explicitly related to our definition of coarse graining, and shows that the training of DNNs with SGD pushes them into becoming progressive coarse-graining machines over time.

Let us operationalise what this perspective means for classification and prediction. In a classification context, we can say that as we go deeper into the network each successive layer has an effective feature space that is lower rank and more predictive of class membership, until the terminal features can be used to perform class prediction via linear regression. In a prediction context, each layer has an effective feature space that is lower rank and more informative about the next time step, until at the final layer the next token or pixel can (again) be predicted using linear regression over the terminal features.

### 4.3 Overparameterisation and Underparameterisation

With this feature space-oriented idea in mind, we can return to the historic debates about overparameterisation and underparameterisation. If the network is underparameterised, the coarse-graining process is stopped prematurely. If the network is overparameterised, generalisation improves as you can coarse-grain beyond special-case features optimised for the training set only into generally predictive features for the problem as a whole. This holds provided the training set is sufficiently representative of the true distribution. From this perspective, the "rank deficiency" phenomenon identified in Feng et al. (2022) indicates that the neural network has coarse-grained too much, and lost too much information. It can no longer keep semantically independent features independent of each other in feature space. Skip operations in residual networks can be seen as a way of combatting such information loss, delaying rank deficiency.

### 4.4 DNNs as Unmixers

The authors of Gutknecht et al. (2025) propose a similar and related hypothesis to the idea of DNNs as progressive coarse-graining systems. They suggest that one of the primary operations of DNNs is in effect a kind of "unmixing", where DNNs take information that has been "mixed" or distributed over multiple data points and recover the information in a way that is linearly separable, and therefore amenable to classification via linear regression. For example, a wing of a parrot might be distributed over 30x30 pixels, and it is the job of the DNN to recover a single "wing" or "wing colour" feature from that distributed mass if it is useful for e.g. bird classification.

They show that as you go deeper into the layers of a neural network, information becomes more redundant and less vulnerable, after giving specific information-theoretic definitions of those terms. In the image classification context, for example, class information is distributed over many sources (pixels) and easily disrupted by manipulation of a few of those sources. We know this thanks to the adversarial attacks which have been developed for CNNs and image transformers which change only relatively few pixels but result in large changes in final classification (see Weng et al. (2023)).

The unmixing operation, then, separates out the irregularly distributed information into a series of semantically meaningful and redundant features. This unmixing process is repeated each layer to make classification easier and easier. The progressive unmixing frame is supported by Feng et al. (2022) and He & Su (2023) —feature spaces with approximately redundant features are naturally of lower rank than feature spaces with linearly independent features. It is also harder to change the direction of output vectors that come from these feature spaces by simply changing a few inputs.

### 4.5 DNNs as Lottery Tickets

We now include for consideration the "lottery ticket hypothesis" proposed by Frankle & Carbin (2019) and extended by Zhou et al. (2020), Malach et al. (2020), Pensia et al. (2021) and various others [6]. The hypothesis by Frankle and Carbin in Frankle & Carbin (2019) states that within a large

---

[6] A survey of recent research is given in Liu et al. (2024).

randomly initialised neural network there is a much smaller sparse sub-network called the "lottery ticket". Training only this subnetwork can get you almost or exactly the same performance as the original, much larger network. Since then various extensions have been made, including the strong hypothesis stated in Zhou et al. (2020) and extended by Malach et al. (2020) and Pensia et al. (2021) which suggests that simply pruning away the rest of the original network (by setting weights to 0) also gives you equivalent performance. The key subnetwork does not even need to be trained, simply setting the sign for each scalar value correctly suffices.

In light of the results we have examined so far this hypothesis seems eminently reasonable: if latter feature spaces are truly filled with redundant and approximately linearly dependent features after training (referring again to Gutknecht et al. (2025)), then large parts of those latter layers can be zeroed out with no impact on the network's ability to find and detect important features. Furthermore, if the rank diminishing is sharp and rapid, especially if you use "rank deficient" components as identified in Feng et al. (2022), then large parts of the entire network can effectively be zeroed out with no loss of intrinsic dimension/information in each layer. This gives a strong reason for why the lottery ticket hypothesis holds for massively overparameterised networks.

Furthermore, if a layer after training contains large amounts of redundant features, it can in principle be converted in a much sparser layer via pruning without loss of function. To do this, you can identify and zero out all features that are approximate linear combinations of other features. Then you can rescale connections to the remaining, linearly independent features based on the strengths of connections to the eliminated features. The precise connection between this kind of rescaling and the pruning techniques used in investigations of the lottery ticket hypothesis is not fully clear, and somewhat beyond the scope of this review. In practice pruning does not always preserve a "clean" set of linearly independent features.

## 5 FORMAL DESCRIPTION OF COARSE GRAINING

Based on the arguments above, we propose the following framework for coarse-graining in trained DNNs, extended from Theorem 2 in Feng et al. (2022).

Suppose that each layer $f_i, i = 1...L$ of network $F$ is almost everywhere smooth and data domain $X$ is a manifold[7], then the following conditions hold at each successive layer $f_i$:

$$\text{Rank}_{\text{eff}}(f_i) < \text{Rank}_{\text{eff}}(f_{i-1}) \tag{3}$$

$$\text{Dim}(f_i) < \text{Dim}(f_{i-1}) \tag{4}$$

Now we add the contributions from Goldfeld et al. (2019). Let $S_i$ be the set of outputs for layer $i$, and $I$ be mutual information. Then we can say

$$I(S_i, \dot{Y})) > I(S_{i-1}, \dot{Y}) \tag{5}$$

$$I(S_i, \dot{X})) < I(S_{i-1}, \dot{X})) \tag{6}$$

Here $\dot{X}$ and $\dot{Y}$ stand for the set of all inputs and labels in the training dataset, not manifolds.

And using the measure of separability $D := \text{Tr}(\text{SS}_w \text{SS}_b^+)$ established in He & Su (2023) [8], we can also point out that

$$D_i > D_{i-1} \tag{7}$$

We call the combination of all of these phenomena *coarse-graining*.

**DEFINITION 1.** Coarse-graining means that each layer $f_i$ produces a **global reduction in complexity** and a **local reduction in complexity** while **increasing classification accuracy**.

---

[7]In accordance with the manifold hypothesis laid out in Cayton (2005)

[8]In words: the data separability metric $D$ for some layer $f_i$ is defined as the trace of the product matrix created by the within-class sum of squares $\text{SS}_w$ and the Moore-Penrose inverse of the between-class sum of squares $\text{SS}_b^+$, with the sum of squares taken over the outputs of $F_{1:i}$ when given every data point for every class.

The **global reduction in complexity** is given by the reduction in intrinsic dimensionality from $f_{i-1}$ to $f_i$ seen in equation (4). Importantly, a global reduction does not imply a uniform reduction. This reduction is not merely blurring an image, for example.

The **local reduction in complexity** can be defined as a reduction in the complexity of the outputs of a DNN layer when compared to the inputs to that layer. Consider the set of all inputs $\dot{X}$. Using $K(\cdot)$ as the Kolmogorov complexity function, we should observe that

$$K(F_{1:i-1}(\dot{X})) > K(f_i(F_{1:i-1}(\dot{X})))  \tag{8}$$

To explain the intuition behind this, we can look at the first layer $f_1$, which takes as input an element of $\dot{X}$ and produces an output which will be a member of $S_1$. For classification and prediction the set of output labels $\dot{Y}$ will simply be one-hot vectors. Therefore, for any reasonable choice of classification or prediction problem $K(\dot{X}) >> K(\dot{Y})$ [9].

We know from Goldfeld et al. (2019) that the following is true:

$$I(S_1, \dot{Y}) > I(\dot{X}, \dot{Y})$$
$$I(S_1, \dot{X}) < I(\dot{X}, \dot{X})$$

This is taken from equations (5) and (6), and from there we can construct the same inequalities between $S_1$ and $S_2$:

$$I(S_2, \dot{Y}) > I(S_1, \dot{Y})$$
$$I(S_2, \dot{X}) < I(S_1, \dot{X})$$

We can also do this for $S_2$ and $S_3$, and so on.

Normally, this formulation indicates that the intermediate representations become less predictive of the inputs and more predictive of the outputs as you go down the layers. Since mutual information is a symmetrical measure, we can also reverse this. In other words, the inputs (which have a long description length) become less predictive of the representation, while the outputs (which have a short description length) become more predictive of the representation. Since we know that the intrinsic dimensionality of each layer is progressively decreasing, we rule out the idea that the intermediate representations are becoming more complex in a way that aligns less well with $\dot{X}$ and more well with $\dot{Y}$.

In addition, the phenomenon of feature redundancy observed in Gutknecht et al. (2025) suggests that more components within the intermediate representations should become correlated with each other, reducing the information value of each feature and therefore of the representation as a whole.

Therefore, we extend the results in Goldfeld et al. (2019) to mean that the description length of the representations is also decreasing, leading to a local decrease in complexity. This formalises the notion from Shi et al. (2025) about "transforming the data to a regular low-dimensional geometry".

Finally, the **increase in classification accuracy** is given by the law of data separation proposed in He & Su (2023) and also by the information bottleneck results.

## 6 CONCLUSION

In this paper, we review multiple lines of research on the behaviour of DNNs on a layer-by-layer basis. We show that there are strong and consistent trends in multiple domains which indicate that neural networks perform some kind of information discarding operation each layer. The information

---

[9]For classification or prediction problems where the inputs are also one-hot vectors (i.e. next-token prediction for LLMs), we can consider $\dot{X}$ as the set of input vectors multiplied by the learned embedding layer. This explains a lot of phenomena around embedding layers being able to substantially encode task-relevant information, as seen in Zhang et al. (2025) and Lu et al. (2021).

that is discarded from the inputs cannot be recovered thanks to the lack of a true inverse for a layer function, and the layers themselves form increasingly low-rank and low intrinsic dimensionality subnetworks.

We propose a unified perspective on what information is being discarded based on Erik Hoel's theory of coarse-graining, where local and context-specific information is discarded for better predictive performance. This allows us to unify multiple parts of the research space, combining insights from novel proposed summary statistics for learning systems, numerical analysis of layers as functions, and information bottleneck theory. The resulting coarse-graining perspective makes empirical predictions about hierarchies of features which have since been proven in research such as Dorszewski et al. (2025), suggesting further lines of research into simple-to-complex feature hierarchies that no longer rely on built in inductive biases of architectures like CNNs.

## 7 ETHICS STATEMENT

After reviewing the ICLR Code of Ethics, I do not believe that this paper contains any potential violations of the Code of Ethics.

## 8 LARGE-LANGUAGE MODEL (LLM) USE STATEMENT

An early draft of the Prior Work section of this paper was reviewed by the Claude LLM. It did not suggest topics that were not brought up by other human collaborators, but did suggest minor fixes and areas to emphasise. These were not copied directly into the paper itself, and since then the paper has received no further LLM contributions.

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

## A APPENDIX: INFORMATION-THEORETIC DESCRIPTION OF OPTIMAL COARSE-GRAINING

Our idea of sacrificing microscale information for macroscale accuracy can also be related to Aaronson et al.'s idea of complextropy, formalised in Aaronson et al. (2014). Specifically, the discarding of "non-necessary", microscale, or irrelevant information in a classification or prediction context can be explained as the principled reduction of information content of the input from $K(x)$ to a value approximating $K(x) - K(x|S)$. Here $x$ is some input, $S$ a system describing a set of elements of which $x$ might be a "generic member", and $K$ is Kolmogorov complexity. With regards to DNN classification, $S$ describes the possible members of a given class, for DNN prediction $S$ describes the possible precursors for some next token or next pixel value. $K(x|S)$ is the description length of $x$ assuming it is a member of $S$, and $K(x|notS)$ the description length of $x$ assuming it is *not* a member.

Here, coarse-graining means that we start with an input whose information content is the Kolmogorov complexity of the input expressed as an $n$-bit string (e.g. a normal picture of a cat), then we progressively discard all the "random" information which separates $x$ from any "generic" member of $S$. Similar to the formulation of mutual information, we find the most apt formation of this intuition to be $K(x) - K(x|S)$. It can also be written $K(x|notS)$. We suggest that this is read as "the length of the program needed to describe $x$, minus the description length of the information distinguishing $x$ from a generic member of $S$". If this value is high, that means that $S$ is a useful description of $x$, such that presuming $notS$ requires you to fill in lots of descriptive information. If the value is low, this means that $S$ is a poor descriptor of $x$, such that removing $S$ from the description of $x$ does not increase the description length of $x$ by much at all. In other words, if we assume that a tabby cat is not a cat, we then have to fill in a lot of information about its legs, fur, whiskers etc., for which the single word "cat" would have been mostly sufficient. If we assume a stone table is not a cat, that shouldn't really change the description of the table at all.

In the language of apparent complexity, each layer of a DNN is like a blurring or denoising function, taking away a part of the incidental or "random" information until all that is left is the "non-random" signal. What remains should be effectively the correlation of $x$ with $S$, or the amount of shared "non-random" information between $x$ and $S$. In the cat classifier case, this would be the amount of "generic-cat" information contained in the image. In the DNN context the amount of shared information is interpreted as the strength of the terminal feature for class $S$, which becomes probability $x$ is a member of class $S$. Here the neural network is used to encode the system $S$, and the discrimination process is equivalent to feeding the input into the neural network.

Importantly, it should not be possible to find more information about the correlation between $x$ and $S$ than $K(x) - K(x|S)$. This is related to the idea of maximising $I(L, Y)$ given in Butakov et al. (2024). Of course, it is trivially possible to discard information beyond this lower bound, but that leads to the rank deficiency phenomenon highlighted in Feng et al. (2022).

