# OpenReview forum: "Progressive Coarse-graining and Deep Neural Networks (DNNs)"
_ICLR.cc/2026/Conference — Submitted to ICLR 2026_

### Official Review · Reviewer_rxiF · 2025-10-23

**Soundness:** 1
**Presentation:** 2
**Contribution:** 1
**Rating:** 2
**Confidence:** 4

**Summary:**

The manuscript is a overview on the working principles of deep neural networks, with a focus on the networks for classification tasks.  The overview is focused on the hypothesis that DNNs perform a progressive coarse-graining of the data, an hypothesis that is for sure viable and relevant.

**Strengths:**

The material described in Section 5 is, to the best of my knowledge, novel.

**Weaknesses:**

1. The perspective ignores a relevant part of the literature. For example the section 4.3, on overparametrization, ignores the fact that the phenomenon is now well understood also analytically, with hundreds of papers devoted to a rigorous analysis of the double descent phenomenon (see for example https://onlinelibrary.wiley.com/doi/pdfdirect/10.1002/cpa.22008 ).
2. Also the hypothesis that DNN perform a progressive reduction of the ranks has been severely challenged by the empirical evidence that in convolutional NNs for image classification the intrinsic dimension does decrease in the last layers, but in the first layers it systematically grows: https://papers.nips.cc/paper_files/paper/2019/file/cfcce0621b49c983991ead4c3d4d3b6b-Paper.pdf
3. I am not sure that conference proceedings are the appropriate venue for reviews and perspectives

**Questions:**

How can the theoretical derivation presented in Section 5 be applied to the analysis of activation of real networks? What would be the added value with respect to the analysis performed in Shwartz-Ziv & Tishby 2017?

---

### Official Review · Reviewer_ALXz · 2025-10-28

**Soundness:** 1
**Presentation:** 1
**Contribution:** 1
**Rating:** 0
**Confidence:** 3

**Summary:**

This paper presents a unifying theoretical perspective, regarding DNNs as a process of "progressive coarse-graining."  The authors synthesize evidence from several lines of recent research, including rank diminution, the information bottleneck, data separation laws, conceptual hierarchies, and the lottery ticket hypothesis. This paper argues that a DNN can be described as a principled simplification process with the goal of procedurally removing irrelevant information from the input, resulting in regular and predictive features that can be used to minimize loss.

**Strengths:**

See weakness

**Weaknesses:**

1. Lack of originality. If I understand correctly, the main core idea of this manuscript, Namely, "A deep neural network can be described as a principled simplification process with the goal of procedurally removing irrelevant information from the input, resulting in ...” This seems to be widely acknowledged, not surprisingly.

2. The justifications for the claims in this paper are very brief. For example,

-*"Therefore, we extend the results in Goldfeld et al. (2019) to mean that the description length of the
representations is also decreasing, leading to a local decrease in complexity. This formalises the
notion from Shi et al. (2025) about ”transforming the data to a regular low-dimensional geometry”."*

-*"Finally, the increase in classification accuracy is given by the law of data separation proposed in
He & Su (2023) and also by the information bottleneck results."*

Without clearly stating the logic, it is difficult for the reader to follow.

3. This paper is composed of previous works and plain claims. **There is no any theoretical proof and experimental verification**

4. Writing logic needs to be polished. It seems that a large portion of this manuscript was generated with LLM assistance. If not, please correct me.

**Questions:**

1. What‘s is Erik Hoel’s Causal Emergence 2.0 framework. This paper mentions it multiple times but does not explain it at all
2. Can the author provide any theoretical proof or experimental verification?

---

### Official Review · Reviewer_zhBV · 2025-10-31

**Soundness:** 1
**Presentation:** 2
**Contribution:** 1
**Rating:** 2
**Confidence:** 3

**Summary:**

This paper suggests an idea: Deep Neural Networks (DNNs) are a kind of progressive coarse-graining system. The authors believe that DNNs improve their high-level prediction by getting rid of small, low-level details layer by layer.

**Strengths:**

The author combined the main ideas from previous papers and summarized them in an intuitive way, including Information Bottleneck Theory, Rank Diminishing,    Hierarchies of Features, Over-parameterization and Under-parameterization, Lottery Ticket Hypothesis.

**Weaknesses:**

1. No new ideas: The paper mostly brings together existing research. While the way it connects these ideas is easy to understand, it doesn't offer any truly new concepts.
2. No proof or experiments: The paper does not provide any mathematical proofs or practical experiments. This makes it hard to know if its ideas are correct.

**Questions:**

Since this paper mostly summarizes existing ideas and doesn’t add new theories, methods, or experiment results, I don’t have specific questions about its content.

---

### Official Review · Reviewer_zCb3 · 2025-11-01

**Soundness:** 2
**Presentation:** 1
**Contribution:** 1
**Rating:** 2
**Confidence:** 5

**Summary:**

This paper presents a conceptual perspective that views deep neural networks as progressive coarse-graining systems.
The authors survey a wide range of existing studies on information bottleneck theory, rank reduction, feature hierarchies, Bayesian interpretations, and the lottery ticket hypothesis.
They argue that these diverse findings can be unified under a single narrative in which each layer of a network reduces effective rank and mutual information with the input while improving class separability with respect to the output.
To formalize this idea, the paper restates known relationships between rank, dimensionality, and mutual information as a set of qualitative inequalities.
The work is primarily conceptual and does not include new theoretical derivations, proofs, or empirical results.

**Strengths:**

The paper is written clearly and covers relevant literature accurately.

**Weaknesses:**

1. The submission does not contain new theoretical or empirical results. It mainly summarizes and rephrases existing ideas.
2. The proposed “progressive coarse-graining” perspective remains qualitative and does not lead to any testable prediction or measurable formulation.
3. Overall, the work reads as a conceptual survey rather than a research contribution.

**Questions:**

1. What specific new understanding or predictive claim is introduced by this framework?
2. How could the proposed perspective be verified or falsified in practice?

---

### Meta-Review · Area_Chair_bevT · 2026-01-06

**Summary:**

The paper proposes a conceptual framework viewing Deep Neural Networks (DNNs) as "progressive coarse-graining" systems. The authors attempt to unify several disparate theories—including Information Bottleneck, rank reduction, feature hierarchies, and the Lottery Ticket Hypothesis.

While the reviewers appreciated the intuitive synthesis of these topics, the recommendation for Reject is unanimous. The primary reason is the lack of technical novelty: the paper neither introduces new theoretical proofs nor provides empirical experiments to validate its claims. Furthermore, reviewers noted that the "coarse-graining" perspective remains qualitative and does not lead to any testable prediction or measurable formulation.

**Reviewer Concerns:**

No rebuttal.

**Reviewer Scores:**

Given the weaknesses (lack of novelty and lack of evidence), it is unlikely that a discussion would have significantly changed the scores.

---

### Decision · Program_Chairs · 2026-01-26

Reject